# Fusion of deterministically generated photonic graph states

Philip Thomas[1], Leonardo Ruscio[1], Olivier Morin[1 ✉] & Gerhard Rempe[1]

Entanglement has evolved from an enigmatic concept of quantum physics to a key ingredient of quantum technology. It explains correlations between measurement outcomes that contradict classical physics and has been widely explored with small sets of individual qubits. Multi-partite entangled states build up in gate-based quantum-computing protocols and—from a broader perspective—were proposed as the main resource for measurement-based quantum-information processing[1,2]. The latter requires the ex-ante generation of a multi-qubit entangled state described by a graph[3–6]. Small graph states such as Bell or linear cluster states have been produced with photons[7–16], but the proposed quantum-computing and quantum-networking applications require fusion of such states into larger and more powerful states in a programmable fashion[17–21]. Here we achieve this goal by using an optical resonator[22] containing two individually addressable atoms[23,24]. Ring[25] and tree[26] graph states with up to eight qubits, with the names reflecting the entanglement topology, are efficiently fused from the photonic states emitted by the individual atoms. The fusion process itself uses a cavity-assisted gate between the two atoms. Our technique is, in principle, scalable to even larger numbers of qubits and is the decisive step towards, for instance, a memory-less quantum repeater in a future quantum internet[27–29].

The characteristics and capabilities of highly entangled graph states[1,4] have been widely explored in theoretical quantum information science. These states form a useful subclass of multi-partite entangled states and have the common feature in that they can be represented by a graph comprising vertices and edges (Fig. 1a). A variety of quantum-information protocols have already been implemented in proof-of-principle experiments with graph states made of entangled photons from spontaneous parametric down-conversion (SPDC) sources[8,30–32]. However, the intrinsically low efficiency of the probabilistic SPDC process remains a substantial obstacle for scalability to large qubit numbers. An alternative and, in principle, deterministic approach using a sequence of single photons emitted from a single memory spin was recognized early on[5,7,33] but could not be realized owing to technological shortcomings. The strategy was implemented only recently but with remarkable progress[9–15] that finally led to an outperformance of SPDC systems in the achievable number of entangled photons[16].

Although these experiments were limited to elementary photonic graph states such as Greenberger–Horne–Zeilinger (GHZ) and linear cluster states (Fig. 1a), several emitter qubits can, in principle, be combined using quantum logic operations to fully exploit their capabilities[17–19,27,28]. Once implemented, this would enable architectures that can generate more complex types of graph state for which many powerful quantum-information protocols such as measurement-based quantum computers and quantum repeaters have been proposed[2,3,6,27,28]. Although recent proposals have successfully identified resource-efficient protocols for such architectures[20,21,34], the emitters still need to satisfy some demanding conditions: a suitable energy-level structure for spin–photon entanglement, efficient emission of indistinguishable photons, coherent control of the emitter qubit and high-fidelity entangling gates between emitters. Despite the individual demonstration of these components, no experiment has yet achieved the successful integration of all of them into the same physical system.

Here we demonstrate fusion of photonic graph states produced from two individually addressable atoms in an optical cavity. First, we implement an atom–atom entangling gate based on two-photon interference in the cavity mode[23,24]. Extending previous work[16], we then show that two graph states separately generated from both emitters can be fused into a larger graph (Fig. 1b). In particular, we demonstrate the generation of two important multi-qubit states, namely ring and tree graph states (see Fig. 1a). Both types of state have been identified as valuable resources for protection against qubit loss and/or computational errors in the framework of measurement-based quantum computation and communication[25–28,35].

Our experimental setup is schematically shown in Fig. 1d and consists of two $^{87}$Rb atoms trapped in a high-finesse optical cavity. Both atoms are positioned at anti-nodes of the cavity mode to ensure strong light–matter coupling with a cooperativity of $C = 1.8$ and hence to enable efficient generation of single photons by means of a vacuum-stimulated Raman adiabatic passage (vSTIRAP)[36]. The cooperativity $C = g^2/(2\kappa\gamma)$ is defined in terms of the cavity quantum electrodynamics parameters $(g, \kappa, \gamma)/2\pi = (5.4, 2.7, 3.0)$ MHz. Here $g$ denotes the coupling rate of a single atom to the cavity mode for the D$_2$ line transition $|F = 1, m_F = \pm 1\rangle \leftrightarrow |F' = 2, m'_F = \pm 2\rangle$, $\kappa$ is the total cavity-field decay rate and $\gamma$ is the atomic-polarization decay rate. We use the atomic-state notation $|F, m_F\rangle$ ($|F', m'_F\rangle$), in which $F$ ($F'$) denotes the total angular

[1]Max-Planck-Institut für Quantenoptik, Garching, Germany. ✉e-mail: olivier.morin@mpq.mpg.de

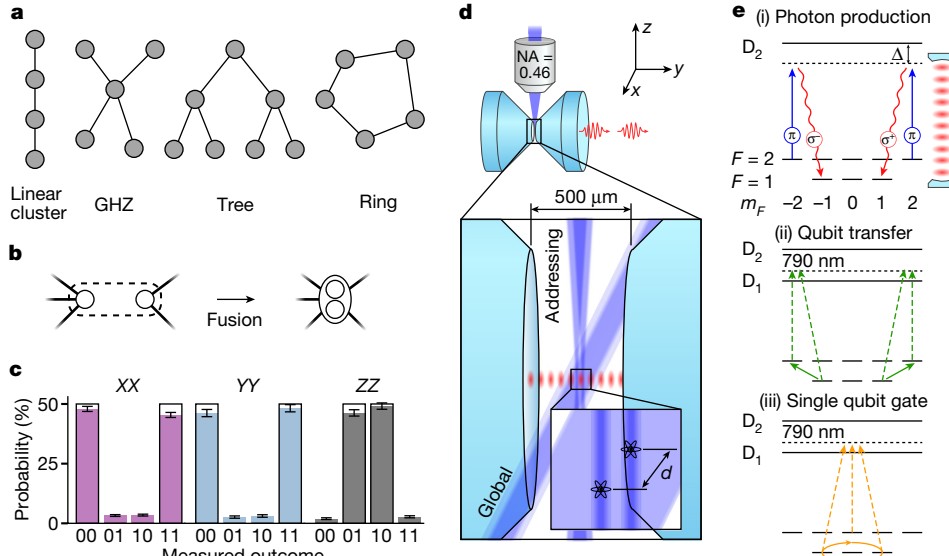

**Fig. 1 | Toolbox for generating photonic graph states. a**, Common examples of graph states. Qubits are represented by vertices, whereas edges connecting them reflect their entanglement topology. **b**, Two independent graphs can be merged by means of a cavity-assisted fusion gate. **c**, Correlation measurements on the Bell state $|\Psi^+\rangle = (|01\rangle_S + |10\rangle_S)\sqrt{2}$. Correlations in the measurement bases *XX*, *YY* and *ZZ* certify entanglement and good agreement with the ideal $|\Psi^+\rangle$ state. Error bars represent the $1\sigma$ standard error. **d**, Experimental apparatus. Two highly reflective mirrors form an asymmetric high-finesse cavity in which

we optically trap two $^{87}$Rb atoms at a distance of $d = (9 \pm 6)$ µm. The vSTIRAP control laser can be applied either globally or atom-selectively using a high-NA objective. In the process, photons are generated leaving the cavity predominantly through the right mirror. **e**, Atomic-level scheme illustrating different steps in the protocol. (i) Photon emission from $|2, \pm2\rangle$. (ii) Before every photon emission, the qubit is mapped from $|1, \pm1\rangle$ to $|2, \pm2\rangle$ using Raman lasers at 790 nm. (iii) The same Raman laser system is used to perform single qubit gates on the qubit states $|1, \pm1\rangle$ (Raman lasers are not shown in panel **d**).

momentum of the ground (excited) state and $m_F$ ($m'_F$) its projection along the quantization axis. The latter is given by a magnetic field oriented along the $y$ axis (cavity axis), giving rise to a Zeeman splitting with Larmor frequency $\omega_L/2\pi = 100$ kHz. To minimize crosstalk between the two emitters, the cavity resonance is detuned by $\Delta/2\pi = -150$ MHz with respect to the $|F = 1, m_F = \pm1\rangle \leftrightarrow |F' = 1, m'_F = \pm1\rangle$ transition[37]. The photons are outcoupled from the cavity and directed towards a polarization-resolving detection setup consisting of a polarizing beam splitter and a pair of single-photon detectors. The vSTIRAP control laser can be applied either globally along the $x$ direction acting on both atoms simultaneously or atom-selectively using an acousto-optic deflector (AOD) combined with a high-NA objective on the $z$ axis. Furthermore, atomic-state manipulation such as optical pumping or coherent driving of Raman transitions can be carried out on both atoms simultaneously using global laser beams.

Thanks to the single-atom addressing beam, both atoms serve as independent emitters, each capable of generating individual spin–photon entanglement in parallel. An atom residing in a coherent superposition of the states $|2, \pm2\rangle$ can undergo a two-photon transition (vSTIRAP) to $|1, \pm1\rangle$, emitting a photon into the cavity mode (see Fig. 1e(i)). We choose the states $|0\rangle_S \equiv |1, +1\rangle$ and $|1\rangle_S \equiv |1, -1\rangle$ as the atomic qubit basis ('S' for 'spin'). In the emission process, conservation of angular momentum gives rise to entanglement between the polarization of the photon and the atomic spin state. $|0\rangle \equiv |R\rangle$ and $|1\rangle \equiv |L\rangle$ define the photonic qubit, with $R/L$ corresponding to right/left circular polarization, respectively. This process can be repeated after a Raman transfer from $|1, \pm1\rangle$ back to $|2, \pm2\rangle$. Using a specifically designed alternating sequence of photon emissions and atomic qubit rotations, elementary graph states such as GHZ or linear cluster states can be obtained[16].

The global beam provides the possibility to entangle the two emitters, thus merging the graphs to which they are connected. The underlying mechanism involves two-photon interference in the cavity mode and resembles the type II fusion gate[38]. Although not strictly identical to fusion in its original form, we here refer to our implementation as a

'cavity-assisted fusion gate'. The quality of this process depends crucially on the indistinguishability of the photons, which is ensured here by both atoms coupling to the same cavity mode and vSTIRAP control laser. To quantitatively characterize this process, we use the cavity-assisted fusion gate to entangle the two atoms and analyse the correlations in the obtained two-qubit state. Hence we initialize both atoms by optical pumping to the state $|2, 0\rangle$. Next, we carry out the fusion by applying a global vSTIRAP control pulse generating two entangled spin–photon pairs. As the photons interfere in the cavity mode, the which-atom information is erased. Therefore, subsequent measurement in the $Z(R/L)$ basis, with one photon being detected in each detector, projects the atoms onto the Bell state $|\Psi^+\rangle = (|01\rangle_S + |10\rangle_S)/\sqrt{2}$ and indicates the success of the entangling operation. A detection of both photons in the same detector projects the atoms onto a product state ($|00\rangle_S$ or $|11\rangle_S$), which means failure.

For the successful preparation of $|\Psi^+\rangle$, we observe strong correlations when measuring the atoms in the bases *XX*, *YY* and *ZZ* (Methods). The probability of each measurement outcome in the different bases is plotted in Fig. 1c. From this, we obtain a state fidelity $\mathcal{F} = 0.915 \pm 0.005$ with respect to the ideal state. This number varies between $0.851 \pm 0.006$ and $0.963 \pm 0.008$, depending on the choice of post-selection criteria for the photon arrival times (Methods). The scenario described above is the simplest case of the fusion mechanism shown in Fig. 1b, in which the emitter qubits do not share a bond with any other qubit before the fusion. The resulting state $|\Psi^+\rangle$ can be interpreted as a logical qubit redundantly encoded[19,38] in the basis $\{|0\rangle_L \equiv |10\rangle_S, |1\rangle_L \equiv |01\rangle_S\}$ ('L' for 'logic'). In the graph-state picture, we express this as a vertex containing two circles. As we will see below, the same principle applies when the two atoms are part of a graph state and do share bonds with other qubits. In this case, the two emitter vertices are merged, preserving the bonds attached to them, as shown in Fig. 1b. If the fusion fails in case both photons end up in the same detector ($RR$ or $LL$), the emitter vertices are removed from the graph. Although this implies a failure of the protocol, the portion of the graph generated up to this point can still be recovered.

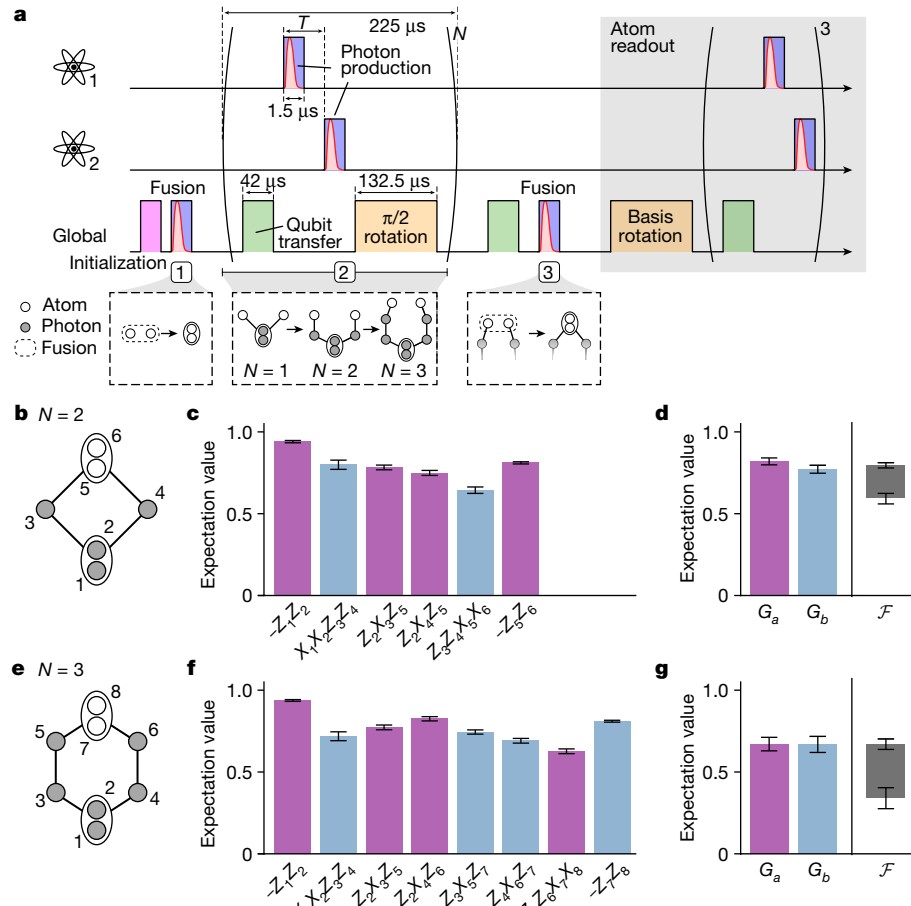

**Fig. 2 | Ring graph states. a**, Experimental protocol showing the individual steps such as initialization (purple), vSTIRAP control pulse (blue), photons (red), qubit transfer (green) and qubit rotation (orange). The colour coding follows Fig. 1e. Each of the three lines represents operations carried out on atom 1, atom 2 or globally, that is, by means of a global laser beam acting on both atoms simultaneously. The multi-qubit quantum state is shown below in its graph representation at different stages of the sequence. The full protocol including atom readout takes 1 ms for the box and 1.2 ms for the hexagon graph

states. **b,e**, Graph representation of the generated 'box' and 'hexagon' graph states. **c,f**, Measured expectation values for the stabilizers corresponding to the box-shaped/hexagon-shaped graphs in **b** and **e**. The colour of the bars (purple and blue) identifies the two sets of stabilizers $a$ and $b$ corresponding to the measurement settings $M_a$ and $M_b$. **d,g**, Entanglement witness measurement obtained from coincidences in the measurement settings $M_a$ and $M_b$. Product correlators $G_a$ (purple) and $G_b$ (blue) and fidelity interval defined by the lower (upper) bound $\mathcal{F}_-$ ($\mathcal{F}_+$) in black. All error bars represent the $1\sigma$ standard error.

We now use the two-atom Bell pair as a starting point for preparing various photonic graph states. As a first example, we demonstrate the generation of ring graph states consisting of up to eight qubits (Fig. 2e). Basically, we first grow a linear cluster state with the emitters at the ends of the chain, which we then fuse to obtain a ring. The generation steps are shown in Fig. 2a using the graph state representation. We start with a two-atom Bell state $|\Psi^+\rangle$, which we obtain from the cavity-assisted fusion. Next, we apply $N$ photon-generation cycles interleaved with $\pi/2$ pulses, resulting in a linear cluster state with the atoms at its ends. In each cycle, we first perform the atomic qubit transfer to $|2, \pm2\rangle$ (see Fig. 1e(ii)). We then generate one photon from each atom (Fig. 1e(i)), applying the control pulse using the addressing system. The two photons are temporally separated by $T \approx 20$ μs to allow enough time for the AOD to direct the addressing beam to the second atom. Afterwards, we perform a $\pi/2$ rotation on the atomic qubits by means of the intermediate state $|2, 0\rangle$ using a sequence of Raman pulses (similar to ref. 16). Each cycle lasts 225 μs and has the effect of adding two photonic qubits to the linear cluster state. In the final step, we perform the qubit transfer followed by a photon-production pulse by means of the global beam. This realizes the fusion on the emitter qubits, effectively merging both ends of the chain. For $N = 2$ and $N = 3$ photon-generation cycles, this produces either a box-shaped or a

hexagon-shaped graph as shown in Fig. 2b,e, respectively. Here, again, the two atoms carry a logical qubit redundantly encoded in $|10\rangle_S$ and $|01\rangle_S$. This specific protocol only produces ring graph states of even parity, that is, an even number of vertices. However, as we show in Methods, ring graph states of odd parity can be obtained simply by adding a global $\pi/4$ rotation after the initial fusion gate. In the following, we will focus on the protocol as presented above and demonstrate the generation of the box and hexagon graphs, consisting of four and six vertices, respectively.

To characterize the experimentally generated state and compare it with the ideal graph state, we measure its corresponding stabilizer operators. The stabilizers to a given graph are defined as $S_i = X_i \prod_{j \in \mathcal{N}_i} Z_j$, in which $\mathcal{N}_i$ is the neighbourhood of vertex $i$. As the cavity-assisted fusion gate produces vertices that are encoded by two physical qubits, we use the concept of 'redundantly encoded graph states'[19]. These are equivalent to regular graph states up to a local unitary transformation on the redundant physical qubits. The stabilizers to the graphs in Fig. 2b,e are shown on the $x$ axis of Fig. 2c,f. To obtain the expectation value of a given stabilizer, we measure coincidences of the corresponding subset of qubits, for which each qubit is detected in either the $Z$ or the $X$ basis. For photonic qubits, the detection basis is set dynamically by means of an electro-optic polarization modulator. The readout of

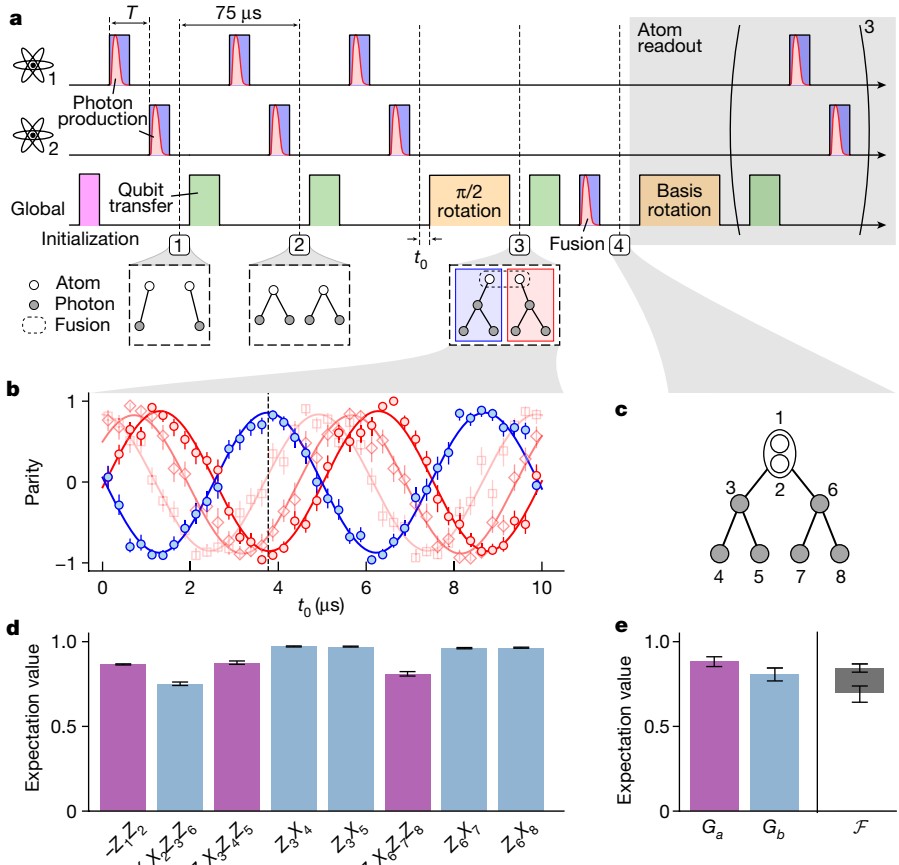

**Fig. 3 | Tree graph states. a**, Experimental protocol similar to Fig. 2a. The length of the individual pulses is the same as for the ring-state protocol. The full protocol including atom readout takes 0.8 ms. **b**, Quantum phase synchronization between the two branches of the tree. The data show the parity of each four-qubit GHZ state as a function of free evolution time $t_0$ before the π/2 rotation. The parity of both branches is tuned to give a relative π phase difference. The dashed vertical line indicates the working point used for the experiment. Data in light red show the parity when slightly varying the photon separation $T$ (see main text). **c**, Graph representation of the generated multi-qubit state with the chosen labelling convention. **d**, Measured stabilizer expectation values similar to Fig. 2. **e**, Product correlators $G_a$ (purple) and $G_b$ (blue) and fidelity interval (black). All error bars represent the 1σ standard error.

the atomic qubit is realized by an appropriate Raman rotation to set the basis, followed by up to three photon-generation attempts with the detection basis set to $R/L$ (Methods).

The experimentally measured expectation values of the stabilizers are shown in Fig. 2c,f for the box and hexagon graphs, respectively. Furthermore, in the case of an even number of vertices, it is possible to divide the stabilizers into two sets $a$ and $b$, which can be measured with two local measurement settings $M_a$ and $M_b$. Similar to in ref. 39, we introduce the operators $G_a$ and $G_b$ as the product $\prod_{i \in a/b} (1 + S_i)/2$ obtained from the measurement setting $M_{a/b}$. Their expectation values can be used to compute the fidelity lower bound $\mathcal{F}_- = \langle G_a \rangle + \langle G_b \rangle - 1$ and upper bound $\mathcal{F}_+ = \sqrt{\langle G_a \rangle \langle G_b \rangle}$ (Methods). Both bounds define a constraint for the fidelity given by the inequality $\mathcal{F}_- \le \mathcal{F} \le \mathcal{F}_+$.

The results are shown in Fig. 2d,g. For the box-shaped graph, we find the fidelity to fall within the interval given by $0.59 \pm 0.03 \le \mathcal{F} \le 0.80 \pm 0.02$. Because the lower bound of this interval exceeds the threshold of 0.5, we have genuine multi-partite entanglement. In the case of the hexagon graph, the data do not prove genuine entanglement, as the lower bound falls below 0.5. Here we have $0.34^{+0.06}_{-0.07} \le \mathcal{F} \le 0.67 \pm 0.03$. We nonetheless observe an overlap with the ideal graph state in terms of the stabilizer expectation values $\langle S_i \rangle$. Also, we emphasize that the true state fidelity is probably higher than the obtained lower bound $\mathcal{F}_-$. Alternative characterization methods have been developed for a more precise estimation of the fidelity[40,41] but are unsuitable for our current detection setup.

As a second example, we demonstrate the generation of a tree graph state[26] consisting of eight qubits (Fig. 3c). In this scenario, we fuse two independent graphs into a larger one. To do this, we first generate two GHZ states, each represented by a star-shaped graph. These will eventually form two branches of the tree graph after merging them using the cavity-assisted fusion gate. The experimental protocol is shown in Fig. 3a. After initialization to the $|2, 0\rangle$ state, each atom emits a photon on successively sending the vSTIRAP control pulse onto the atoms using the addressing system, generating two atom–photon entangled pairs (Fig. 3a, (1)). Two further photon-production cycles are carried out, each cycle consisting of a global Raman transfer to $|2, \pm 2\rangle$ and a photon-generation pulse (Fig. 3a, (2,3)). Next, after a free evolution time of $t_0$, a π/2 rotation is applied to both atomic qubits simultaneously. At this stage, two GHZ states of the form

$$(|+\rangle_S |0 + +\rangle + e^{-i\phi_{1,2}(t_0)} |-\rangle_S |1 - -\rangle)/\sqrt{2} \tag{1}$$

have been generated, each consisting of one atom and three photons. Here $|\pm\rangle = (|0\rangle \pm |1\rangle)/\sqrt{2}$. Note that two of the photons in the above expression have experienced a Hadamard rotation, which—in the experiment—is absorbed into the setting of the measurement basis. This as well as a π/2 pulse on the atomic qubit has the effect that the respective qubit is 'pushed out' from the graph, thus forming a so-called 'leaf' qubit (see, for example, ref. 19).

The second term in equation (1) carries a phase factor $\phi_{1,2}(t_0)$, which arises from the free evolution of atoms 1 and 2, as denoted by the subscript. We write $\phi_1(t_0) = 2\omega_1 t_0$ and $\phi_2(t_0) = 2\omega_1(t_0 - T)$ as functions of the $\pi/2$ pulse timing given by $t_0$, in which $\phi_2(t_0)$ contains the photon separation $T$ as an extra parameter. Figure 3b shows the parity of each of the GHZ states and the oscillating behaviour as a function of $t_0$. We show that, by adjusting $t_0$ and $T$, we can actively tune this phase to be 0 and $\pi$ for the two branches, respectively.

In the next step, the two branches are fused into a tree graph. To do this, we apply a global vSTIRAP control pulse, leading to the simultaneous emission of two photons that are detected in the $R/L$ basis. As before, the protocol succeeds if one photon is detected in each detector, that is, in the $R$ and $L$ polarization states, respectively. This step can be thought of as a projection of the atomic qubits on the subspace $\{|01\rangle_S, |10\rangle_S\}$, given by the operator $|01\rangle_S\langle 01|_S + |10\rangle_S\langle 10|_S$. We then obtain the state

$$|\psi_{\text{tree}}\rangle = \frac{1}{2\sqrt{2}}[|10\rangle_S(|0++\rangle + |1--\rangle)^{\otimes 2} + |01\rangle_S(|0++\rangle - |1--\rangle)^{\otimes 2}], \quad (2)$$

which is an eigenstate to a set of stabilizers corresponding to a tree graph state of depth two, in which the root qubit can be seen as redundantly encoded (Fig. 3c). Owing to the redundant encoding, we again use the modified stabilizers $S_1 = -Z_1 Z_2$ and $S_2 = X_1 X_2 Z_3 Z_6$ for the physical qubits of the root vertex. If necessary, the atoms can be disentangled from the photonic state by performing an atom-to-photon state transfer[16]. In certain cases, however, the protocol may require the emitters to be part of the graph. An example is the one-way quantum repeater[28], in which an emitter forms the root qubit of a tree graph.

The measured expectation values for the stabilizers are shown in Fig. 3d. We find all stabilizers to be above 0.7 and some of them close to 1, in good agreement with the ideal state for which $\langle S_i \rangle = 1$. Furthermore, we are able to prove genuine multi-partite entanglement by collecting eight-qubit coincidences. We find that the entanglement fidelity is constrained by the upper and lower bounds with $0.69^{+0.04}_{-0.05} \leq \mathcal{F} \leq 0.85^{+0.02}_{-0.03}$, thus exceeding the classical threshold of 0.5.

The fidelities of the generated entangled states are limited by various sources of error. For single-emitter protocols, we have identified spontaneous scattering in the photon-emission process and imperfect Raman rotations as the main error mechanisms[16]. In this work, we attribute most of the infidelity to the cavity-assisted fusion gate, which is affected by spontaneous scattering as well as imperfect photon indistinguishability. A more detailed discussion can be found in Methods.

The generation of the presented graph states relies on a high overall source-to-detector efficiency, which—in this work—is close to 0.5 for a single photon emission. Hence, with a coincidence rate on the order of one per minute, we can collect hundreds of events in a few hours of measurement (Methods).

In conclusion, we have generated ring graph states of up to 6 (8) logical (physical) qubits and a tree graph state made up of 7 (8) logical (physical) qubits by coupling two emitters by means of a cavity-assisted fusion gate. The latter constitutes the, in our view, decisive step towards scalable architectures of coupled single-photon sources for creating arbitrary photonic graph states. These could be realized with several atom–cavity systems that are embedded in a distributed architecture and connected by optical fibre links[3]. Alternatively, one could increase the number of emitters within the same cavity, for instance making use of arrays of optical tweezers. Both approaches are conceptually similar, whereas the latter takes advantage of hosting several emitters in the same hardware device. A larger number of emitters would enable tree states of higher depths or repeater graph states, which are proposed as useful tools to overcome photon loss in long-distance transmission lines[6,27,28]. Similar schemes can be used to generate two-dimensional cluster states to enable fault-tolerant quantum-computing protocols such as one-way or fusion-based quantum computation[2,25,35]. Finally, yet notably, the photons of the graph state could be individually steered to and stored in a distributed set of heralded quantum memories[42], thereby bringing the flying entanglement to a standstill in a material system. In the context of multi-partite quantum networks[43], this approach would offer many fascinating possibilities[29] beyond those of a two-party quantum-communication link.

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

# Methods

### Experimental setup

The apparatus used in our work consists of a single-sided high-finesse cavity in which we optically trap two rubidium atoms. Most experimental details about the setup including the cavity quantum electrodynamics parameters have already been described elsewhere[16]. In the following, we provide further information that is important for the current work.

The atoms are trapped in a two-dimensional optical standing-wave potential formed by two pairs of counter-propagating laser beams. The first is a retro-reflected laser at a wavelength of $\lambda = 1{,}064$ nm along the $x$ axis. The second propagates inside the cavity mode along the $y$ axis with $\lambda = 772$ nm. The atoms are loaded from a magneto optical trap to the cavity centre using a second 1,064-nm running-wave laser. The light scattered by the atom during laser cooling is imaged by means of the objective onto an electron-multiplying charge-coupled device camera to spatially resolve the position of the atoms. After each loading attempt, we find a random number of atoms $n$ at random positions. The experimental control software identifies atom pairs with a suitable relative distance $d$. If no such atom pair is present, a new loading attempt starts immediately. Otherwise, a tightly focused resonant laser beam, propagating through the objective and steered by the AOD, removes the $n - 2$ unwanted atoms. The $x$ component of the centre-of-mass position of the atom pair $(x_2 + x_1)/2$ is then actively stabilized to the centre of the cavity mode by acting on the relative phase of the 1,064-nm counter-propagating laser beams. The $y$ components $y_1$ and $y_2$ are controlled by modulating the optical power of the 772-nm intra-cavity trap until the atoms are found in a desired position.

### Fusion gate and post-selection

For a fusion gate to be successful, two photons have to be detected, as described in the main text. Mathematically, this can be understood by considering two atom–photon entangled states of the form

$$|\psi_{AP}\rangle = \frac{1}{\sqrt{2}}\left(|F = 1, m_F = 1\rangle\,|L\rangle - |F = 1, m_F = -1\rangle\,|R\rangle\right). \qquad (3)$$

The relative minus sign in the above equation arises from the Clebsch–Gordan coefficients in the two emission paths. Applying the projector $\langle R|\langle L|$ to the product state $|\psi_{AP}\rangle \otimes |\psi_{AP}\rangle$ corresponds to the detection of an $R$ and an $L$ photon, signalling a successful fusion. This leaves us with the $|\Psi^+\rangle$ Bell state. Here we implicitly assumed that the two photons occupy the same spatiotemporal mode function. In the experiment, however, their temporal wave packet may not be perfectly indistinguishable, leading to an incomplete erasure of which-path information. Such imperfection can arise from spontaneous scattering by means of the excited state or from unbalanced atom–cavity or atom–laser coupling. This effect becomes visible when post-selecting on the arrival time of the photons. The influence of the arrival time on the fidelity of the atom–atom Bell state is summarized in Extended Data Fig. 1. Panel a shows the intensity profile of the photon temporal wave function as a function of $t_{R,L}$, with $t_R$ and $t_L$ being the arrival times of the $R$-polarized and $L$-polarized photons produced in the fusion process, respectively. Events in which a photon arrives outside the time interval marked by the dashed lines are discarded. This interval contains about 98% of all single-photon counts. Panel b is a two-dimensional density plot of the number of two-photon events versus arrival times $t_R$ and $t_L$. We can see that most events lie in the vicinity of the point $t_R = t_L = 200$ ns. The dashed line encloses the region defining the post-selection criteria, which we specify in more detail below. Panel c is a density plot similar to b showing the fidelity as a function of $t_R$ and $t_L$. We find that the fidelity is highest near the diagonal of the plot, that is $t_R \approx t_L$. This motivates our choice of the post-selection

region enclosed by the dashed line. Pixels for which we did not acquire enough data to compute the fidelity are shown in white. The fidelity is computed using the formula

$$\mathcal{F} = \frac{1}{4}(1 + \langle XX \rangle + \langle YY \rangle - \langle ZZ \rangle). \qquad (4)$$

Here $XX$, $YY$ and $ZZ$ are two-qubit operators consisting of the respective Pauli operators. In panels d–f, we analyse their expectation values $\langle XX \rangle$, $\langle YY \rangle$ and $\langle ZZ \rangle$ as a function of arrival time difference $|t_R - t_L|$. We plot the expectation value both for $|t_R - t_L| = \tau$ (orange) and $|t_R - t_L| \leq \tau$ (purple), that is, the cumulative expectation value. We find all correlators to be in good agreement with the ideal case, for which we expect $\langle XX \rangle = \langle YY \rangle = 1$ and $\langle ZZ \rangle = -1$. The high fidelity of the two-atom Bell state is also an indicator of a high photon indistinguishability. The dashed lines indicate the maximum value of $\tau$, that is, $|t_R - t_L| \leq \tau$, chosen for the data presented in Fig. 1c.

**Post-selection criteria.** For the data in Extended Data Fig. 1c, as well as the data presented in the main text, we apply two post-selection steps. The first step consists of restricting the absolute detection time of the photons to a predefined interval of 1 µs width (see dashed lines in Extended Data Fig. 1a). This step applies to both single-photon and two-photon events. The second post-selection condition involves the relative arrival time difference $|t_R - t_L|$ in the case of two-photon events and therefore only applies to photons generated in the fusion process. The diagonal dashed lines in Extended Data Fig. 1b,c mark the condition $|t_R - t_L| \leq \tau_{max} = 250$ ns. Events in which the photons are detected with a relative delay larger than $\tau_{max}$ are discarded. In about 80% of experimental runs, the two photons fall within the interval of $\tau_{max}$.

As stated in the main text, the atom–atom Bell-state fidelity ranges between 0.851(6) and 0.963(8). The first number refers to the scenario in which no post-selection on the photon arrival time is applied. The second number is obtained when restricting the photon arrival times to $t_{R,L} \leq 500$ ns and $|t_R - t_L| \leq 20$ ns. In this case, the post-selection ratio is about 15%.

The above numbers refer to the scenario in which the atom is initialized to $|F = 2, m_F = 0\rangle$ before photon generation. However, in the ring and tree states protocol, the last fusion step consists of a two-photon emission from $|F = 2, m_F = \pm 2\rangle$. In this case, the photon wave packet is slightly longer, as the $m_F = \pm 2$ Zeeman sublevels couple to different excited states in the emission process. Here we apply the same 1-µs time interval as for the $m_F = 0$ case, as at least 95% of the photon wave packet is enclosed by this window. However, for the two-photon events in the fusion process, we choose a maximum time difference of $\tau_{max} = 400$ ns to accommodate for a post-selection fraction of about 80%, similar to the $m_F = 0$ case.

### Atom readout

At the end of the generation sequence for tree and ring graph states, the atomic qubits are still entangled with the photons previously generated. One way to measure the atomic qubits is to perform an atom-to-photon state transfer, as done in ref. 16. Here the qubit is mapped from $|F = 1, m_F = \pm 1\rangle$ to $|F = 2, m_F = \pm 1\rangle$ before photon production. In this way, the qubit is fully transferred to the photon, which can then be measured optically. In this work, however, we chose another technique to measure the atomic qubit. For a $Z$ measurement, we transfer the qubit to $|F = 2, m_F = \pm 2\rangle$ and generate a photon measuring it in the $R/L$ basis. Detecting an $R$ ($L$) photon projects the atomic qubit onto the state $|0\rangle_S$ ($|1\rangle_S$). When measuring the qubit in $X$ or $Y$, we set the basis directly on the atomic qubit with a π/2 pulse whose phase is tuned according to the basis. The advantage of this scheme is that it can be repeated until success in the case of photon loss, thus increasing the overall efficiency of the state readout. However, as errors are

more likely to occur after many repetitions, we limit the number of attempts to three.

## Detailed protocol description

In the following, we will describe the generation protocol for the ring and tree graph states with explicit expressions for each step. In the derivation, we do not explicitly include the free evolution of the atomic qubit. In the experiment, the phases that arise from the qubit oscillation are tracked by measuring the stabilizer operators as a function of certain timing parameters related to, for instance, Raman transfers and photon emissions. Notably, these phases may be tuned for each atom independently by varying the respective time of the photon-production pulse.

**Ring states.** We first describe the protocol of the ring graph states and choose the pentagon ring as a specific example. The box-shaped and hexagon-shaped graphs are obtained from a similar protocol, only omitting a single π/4 rotation. A sketch of the experimental sequence is given in Extended Data Fig. 2a.

The first step of the protocol is to entangle the two atoms and prepare them in the Bell state $|\Psi^+\rangle = (|01\rangle_S + |10\rangle_S)/\sqrt{2}$. To obtain the pentagon graph, which has an odd number of vertices, we need to apply a global $-\pi/4$ pulse. This 'pushes' the two qubits apart, forming two separate vertices with an edge between them (Extended Data Fig. 2a, (2)). The corresponding state (omitting normalization constants) reads

$$\xrightarrow{-\pi/4} |00\rangle_S + |01\rangle_S + |10\rangle_S - |11\rangle_S$$
$$= |0+\rangle_S + |1-\rangle_S. \tag{5}$$

Here we have substituted the transformations

$$|0\rangle_S \xrightarrow{-\pi/4} \cos\left(\frac{\theta}{2}\right)|0\rangle_S + \sin\left(\frac{\theta}{2}\right)|1\rangle_S,$$
$$|1\rangle_S \xrightarrow{-\pi/4} -\sin\left(\frac{\theta}{2}\right)|0\rangle_S + \cos\left(\frac{\theta}{2}\right)|1\rangle_S \tag{6}$$

and used $\theta = -\pi/4$, as well as $\cos\left(\frac{\pi}{8}\right) = \frac{\sqrt{2+\sqrt{2}}}{2}$ and $\sin\left(\frac{\pi}{8}\right) = \frac{\sqrt{2-\sqrt{2}}}{2}$. Subsequently, each atom emits a photon, giving

$$\xrightarrow{PP} |0\rangle_S|0\rangle(|0\rangle|0\rangle_S + |1\rangle|1\rangle_S) + |1\rangle_S|1\rangle(|0\rangle|0\rangle_S - |1\rangle|1\rangle_S)$$
$$= (|0\rangle_S|0\rangle + |1\rangle_S|1\rangle)|0\rangle|0\rangle_S + (|0\rangle_S|0\rangle - |1\rangle_S|1\rangle)|1\rangle|1\rangle_S, \tag{7}$$

followed by a π/2 rotation on the atomic qubits:

$$\xrightarrow{\pi/2} (|+\rangle_S|0\rangle + |-\rangle_S|1\rangle)|0\rangle|+\rangle_S + (|+\rangle_S|0\rangle - |-\rangle_S|1\rangle)|1\rangle|-\rangle_S$$
$$= (|0\rangle_S|+\rangle + |1\rangle_S|-\rangle)|0\rangle|+\rangle_S + (|0\rangle_S|-\rangle + |1\rangle_S|+\rangle)|1\rangle|-\rangle_S, \tag{8}$$

which is equal to a four-qubit linear cluster state with the atoms at both ends of the chain. Note that the (global) π/2 pulse affects only the spin component of the multi-qubit state. We perform another photon production on both spins and obtain

$$\xrightarrow{PP} (|0\rangle_S|0+\rangle + |1\rangle_S|1-\rangle)|0\rangle(|0\rangle|0\rangle_S + |1\rangle|1\rangle_S)$$
$$+ (|0\rangle_S|0-\rangle + |1\rangle_S|1+\rangle)|1\rangle(|0\rangle|0\rangle_S - |1\rangle|1\rangle_S). \tag{9}$$

We apply a $Z$ gate to qubit 6 and a Hadamard to qubits 1 and 6 (indices run from left to right).

$$\xrightarrow{H_1 \otimes Z_6 \otimes H_6} (|+\rangle_S|0+\rangle + |-\rangle_S|1-\rangle)|0\rangle(|-\rangle|0\rangle_S + |+\rangle|1\rangle_S)$$
$$+ (|+\rangle_S|0-\rangle + |-\rangle_S|1+\rangle)|1\rangle(|+\rangle|0\rangle_S + |-\rangle|1\rangle_S). \tag{10}$$

Now we perform the fusion operation on qubits 1 and 6 and obtain

$$\xrightarrow{Fusion} \frac{1}{2\sqrt{2}}(|10\rangle_S(|0+0+\rangle + |1-0+\rangle + |0-1-\rangle + |1+1-\rangle)$$
$$+ |01\rangle_S(|0+0-\rangle - |1-0-\rangle + |0-1+\rangle - |1+1+\rangle))$$
$$= \frac{1}{2\sqrt{2}}(|0\rangle_L(|0+0+\rangle + |1-0+\rangle + |0-1-\rangle + |1+1-\rangle)$$
$$+ |1\rangle_L(|0+0-\rangle - |1-0-\rangle + |0-1+\rangle - |1+1+\rangle)). \tag{11}$$

Here we have moved the second spin qubit to the first position and reintroduced the logical qubit encoding using $|0\rangle_L$ and $|1\rangle_L$. Furthermore, we have added a normalization constant. The above expression represents the state that corresponds to the graph shown in Extended Data Fig. 2c. The measured stabilizer expectation values are shown in Extended Data Fig. 2b.

**Tree states.** We now describe the experimental protocol for generating the target state of the form

$$|\psi_{tree}\rangle = \frac{1}{2\sqrt{2}}[|0\rangle(|0++\rangle + |1--\rangle)^{\otimes 2} + |1\rangle(|0++\rangle - |1--\rangle)^{\otimes 2}]. \tag{12}$$

We start by preparing both atoms in the $|F = 2, m_F = 0\rangle$ state, followed by three sequential photon-production cycles on each atom in parallel. From this, we obtain the tensor product of two GHZ states, each consisting of one atom and three photons (see also ref. 16). Omitting normalization constants, we can write the state as

$$|\psi\rangle = (|0\rangle_S|000\rangle + |1\rangle_S|111\rangle) \otimes (|0\rangle_S|000\rangle - |1\rangle_S|111\rangle). \tag{13}$$

Note that the second term carries a relative minus sign with respect to the first term. This is reflected in the parity measurement shown in Fig. 3b. We now perform a Hadamard gate on all qubits except qubits 2 and 6 (indices run from left to right) and obtain

$$\longrightarrow (|+\rangle_S|0++\rangle + |-\rangle_S|1--\rangle) \otimes (|+\rangle_S|0++\rangle - |-\rangle_S|1--\rangle). \tag{14}$$

For the atoms, the Hadamard is carried out with a Raman laser (see Fig. 1e), whereas for the photons, it is absorbed into the setting of the measurement basis.

We now merge both branches into one larger graph state by applying the fusion gate. Hence we generate two photons from the atoms with the global STIRAP control laser. Detecting one photon in $R$ and one in $L$ effectively projects the atoms onto the subspace $\{|01\rangle_S, |10\rangle_S\}$.

$$\xrightarrow{|01\rangle_S\langle 01|_S + |10\rangle_S\langle 10|_S} (|10\rangle_S + |01\rangle_S)|0++\rangle^{\otimes 2}$$
$$+ (|10\rangle_S - |01\rangle_S)|0++\rangle|1--\rangle$$
$$+ (|10\rangle_S - |01\rangle_S)|1--\rangle|0++\rangle + (|10\rangle_S + |01\rangle_S)|1--\rangle^{\otimes 2}$$
$$= |10\rangle_S(|0++\rangle + |1--\rangle)^{\otimes 2} + |01\rangle_S(|0++\rangle - |1--\rangle)^{\otimes 2}. \tag{15}$$

For convenience, we have moved the second spin qubit to the first position in the above expression, which allows us to express the two atoms as a logical qubit encoded in the basis $\{|0\rangle_L \equiv |10\rangle_S, |1\rangle_L \equiv |01\rangle_S\}$. Adding a normalization constant, we can then write the final state as

$$|\psi_{tree}\rangle = \frac{1}{2\sqrt{2}}[|0\rangle_L(|0++\rangle + |1--\rangle)^{\otimes 2} + |1\rangle_L(|0++\rangle - |1--\rangle)^{\otimes 2}]. \tag{16}$$

This is equal to the expression in equation (12), with the only difference being that the root qubit is now redundantly encoded by the two atoms. Alternatively, it would be possible to remove one of the atoms from the state by an $X$ basis measurement.

## Coincidence rate

For each multi-qubit state, the typical generation and detection rate is between 0.4 and 2.3 coincidences per minute. The total number of events as well as the total measurement time are summarized in Extended Data Table 1 for each graph state generated in this work. These numbers include all post-selection steps as described above.

## Entanglement witness and fidelity bounds

To quantify the agreement between the experimentally produced multi-photon state and the target state, we use an entanglement witness. This has the advantage that we can derive a lower bound of the fidelity without the need for full quantum-state tomography. The fidelity of a density matrix $\rho$ with respect to the target state $|\psi\rangle$ is defined as

$$\mathcal{F} = \text{Tr}\{\rho \ |\psi\rangle\langle\psi|\}. \tag{17}$$

Using the stabilizers, we can express the projector to the target state as

$$|\psi\rangle\langle\psi| = \prod_i \frac{1+S_i}{2} = \prod_{i\in a} \frac{1+S_i}{2} \prod_{j\in b} \frac{1+S_j}{2} = G_a \cdot G_b. \tag{18}$$

Here we have written the projector as a product of two terms $G_a$ and $G_b$ associated with two sets of stabilizers $a$ and $b$. Each set $a/b$ can be measured with a single local measurement setting $M_a/M_b$. These only involve measurements in the $X$ or $Z$ basis for every qubit. We can then write the projector in terms of $G_a$ and $G_b$, giving

$$|\psi\rangle\langle\psi| = G_a \cdot G_b = G_a + G_b - 1 + (1 - G_a)(1 - G_b) \tag{19}$$

As the stabilizers $S_i$ take the values +1 or −1, the product terms $G_a$ and $G_b$ are either 1 or 0. We conclude that $(1 - G_a)(1 - G_b)$ is non-negative. Omitting this term, we find the lower bound

$$\mathcal{F}_- \equiv \langle G_a\rangle + \langle G_b\rangle - 1 \le \mathcal{F}. \tag{20}$$

The above expression is applicable if the stabilizers can be divided into two sets $a$ and $b$, each of which can be measured with a single measurement setting ($M_a$ and $M_b$). In the context of our experiment, this applies to tree graph states as well as ring graph states of even parity, that is, an even number of vertices. To the best of our knowledge, there is no equivalent method for ring graph states of odd parity, such as the pentagon graph, and a fidelity lower bound cannot be derived.

We can further derive a fidelity upper bound based on the terms $G_a$ and $G_b$. First, for any pure state $|\psi\rangle$, we have

$$\langle\psi| \ G_a G_b \ |\psi\rangle \le \sqrt{\langle\psi| \ G_a G_a^\dagger \ |\psi\rangle\langle\psi| \ G_b^\dagger G_b \ |\psi\rangle}, \tag{21}$$

by direct application of the Cauchy–Schwarz inequality. The terms $(1 + S_i)/2$ are projectors, because $S_i^2 = 1$ and therefore

$$\left(\frac{1+S_i}{2}\right)^2 = \frac{1+2S_i+S_i^2}{4} = \frac{1+S_i}{2}. \tag{22}$$

By construction, the stabilizers $S_i$ commute and therefore the projectors $(1 + S_i)/2$ commute as well. Hence, because $G_{a/b}$ are products of commuting projectors, $G_a$ and $G_b$ themselves are also projectors:

$$G_{a/b}^2 = \left(\prod_{i\in a/b} \frac{1+S_i}{2}\right)^2 = \prod_{i\in a/b} \left(\frac{1+S_i}{2}\right)^2 = \prod_{i\in a/b} \frac{1+S_i}{2} = G_{a/b}. \tag{23}$$

Equation (21) can then be simplified as $\langle\psi|G_a G_b|\psi\rangle \le \sqrt{\langle\psi|G_a|\psi\rangle\langle\psi|G_b|\psi\rangle}$.

Then, to generalize to mixed states, we write the mixed state $\rho$ as a linear combination of pure states, that is, $\rho = \sum_k p_k \ |\psi_k\rangle\langle\psi_k|$, and apply the above inequality to each of them:

$$\langle G_a G_b\rangle = \sum_k p_k\langle\psi_k|G_a G_b|\psi_k\rangle \le \sum_k p_k\sqrt{\langle\psi_k|G_a|\psi_k\rangle\langle\psi_k|G_b|\psi_k\rangle}. \tag{24}$$

We identify the right term as a scalar product of two vectors and again use the Cauchy–Schwarz inequality

$$\sum_k \sqrt{p_k\langle\psi_k| \ G_a \ |\psi_k\rangle} \sqrt{p_k\langle\psi_k| \ G_b \ |\psi_k\rangle}$$
$$\le \sqrt{\left(\sum_k p_k\langle\psi_k| \ G_a \ |\psi_k\rangle\right)\left(\sum_{k'} p_{k'}\langle\psi_{k'}| \ G_b \ |\psi_{k'}\rangle\right)}, \tag{25}$$

which shows the upper bound of the fidelity

$$\mathcal{F} = \langle G_a G_b\rangle \le \sqrt{\langle G_a\rangle\langle G_b\rangle} \equiv \mathcal{F}_+. \tag{26}$$

In the next section, we will use both fidelity bounds for a comparison between the experimental data and the expected fidelity.

## Estimation of errors

In our previous work[16], we identified some error mechanisms present in our system. For single-emitter protocols, the main error sources are spontaneous scattering in the photon-emission process (about 1% per photon) and imperfect Raman rotations (about 1% per π/2 pulse). In the following, we discuss several more mechanisms that could negatively affect the fidelity. In some cases, the effect of these mechanisms on the fidelity of multi-qubit entangled states is difficult to quantify because of the complexity of the entanglement topology and the protocols to generate it. Furthermore, measuring the fidelity of multi-qubit states is a non-trivial task and our measurement setup only allows us to extract a lower and an upper bound of the fidelity.

**Fusion gate.** For the two-emitter protocols developed in this work, the cavity-assisted fusion gate is probably the largest source of error. As shown in the main text, this mechanism can be used to prepare the $|\Psi^+\rangle$ Bell state with a fidelity ranging between 0.85 and 0.96, depending on how strictly we post-select on the arrival time of the photons. The fact that the fidelity decreases with a larger arrival time difference $\tau$ (see Extended Data Fig. 1) can be explained by an imperfect indistinguishability of the photons involved in the fusion process. For the standard value of $\tau_{\max} = 250$ ns, the fidelity of the $|\Psi^+\rangle$ Bell state is 0.92. This number includes state readout of the two atoms, each of which is expected to introduce an error similar to a photon emission (roughly 1%). We conclude that the infidelity from the fusion process is on the order of 6%.

**Decoherence.** Another potential source of infidelity is atomic decoherence caused by magnetic-field noise or intensity fluctuations of the optical-trapping beams. We have measured the coherence time of the atomic qubit $T_2$ to be approximately 1 ms. However, the atomic qubit is largely protected by a dynamical decoupling mechanism that is built into the protocol[16], thereby extending the coherence time. The exact extent to which this mechanism takes effect depends on the specific timing parameters in the sequence and the frequency range in which the noise sources are most dominant (for example, magnetic-field fluctuations). Therefore, it is difficult to quantify how much the decoherence translates into infidelity of the final graph state. Furthermore, different types of graph state are more or less susceptible to noise[44]. It is therefore not straightforward to theoretically model the role of decoherence in the fidelity of the final multi-partite entangled state.

**Qubit leakage.** During the protocol, the emitter qubits are continuously transferred between different atomic states. These states are

$|1, ±1⟩$, $|2, ±2⟩$ and $|2, 0⟩$, in which we again write the state as $|F, m_F⟩$ with the quantum numbers $F$ and $m_F$. However, there seems to be a low probability that, during the emission process, the atom undergoes a transition to $|1, 0⟩$ (instead of $|1, ±1⟩$). This is readily explained by and consistent with the finding of spontaneous scattering during the vSTIRAP process, but may equally result from a contamination of $σ^+/σ^-$ polarization components in the vSTIRAP control laser. The latter is in turn caused by either an imperfect polarization setting or longitudinal polarization components owing to the tight focus of the beam. The unwanted $σ^+/σ^-$ components couple to the $|F' = 1, m'_F = ±1⟩$ states and can thus drive a two-photon transition to $|F = 1, m_F = 0⟩$. This process results in the atom leaving the qubit subspace but, unfortunately, such an event remains undetected. If the protocol resumes with a Raman π/2 pulse, the parasitic population in $|1, 0⟩$ is then partly transferred to $|2, ±1⟩$, as the corresponding transitions have the same resonance frequency. A subsequently emitted photon will then yield a random measurement outcome, which is detrimental to the fidelity of the state.

The leakage mechanism described above is difficult to quantify, mainly because our experiment lacks an $m_F$-selective state readout. We do however estimate that the longitudinal polarization components of the addressing beam have a relative amplitude on the order of about 1%, contributing to each single-photon emission. For the global beam, this effect is negligible owing to a larger focus.

**Other sources of error.** Other sources of error include drifts of the optical fibres, such as for the Raman beam, the global and addressing vSTIRAP beam or the optical traps, as well as the magnetic field. Furthermore, the position of the atoms is not fixed but varies from one loading attempt to another. In this work, we chose position criteria that are less strict than those in ref. 16, to increase the data rate. In combination with the drifts mentioned above, this leads to a variance in coupling between the atoms and the cavity, as well as the atoms and different laser beams. As a consequence, this may affect the fidelity of different processes, such as the fusion gate or Raman transfers. Furthermore, a drift of the magnetic field or the light shift induced by the optical trap can influence the phase of the atomic qubits at different stages of the protocol.

A way to reduce the overall infidelity would be to increase the cooperativity $C$. This would reduce the effect of spontaneous scattering, improve photon indistinguishability and thereby increase the fidelity of the fusion process and partly mitigate the qubit-leakage error. Photon emission through the $D_1$ line of rubidium would have a similar effect, owing to a larger hyperfine splitting in the $5^2P_{1/2}$ excited state. Another strategy to improve the system would be a better control of the atom positions by using more advanced trapping techniques, such as optical tweezers. This would greatly reduce all errors associated with the variance of the atom positions. It would also allow longer trapping times and therefore higher data rates.

## Error model

As an (oversimplified) ansatz to estimate the combined effect of the error mechanisms described above, we write the density matrix as a mixture of the ideal density matrix and white noise. This is a common approach to investigate, for instance, the robustness of entanglement witnesses against noise (see, for example, ref. 45). The density matrix then reads

$$\rho_{exp} = (1 - p_{noise})\rho_{ideal} + p_{noise}\frac{1}{2^n}, \qquad (27)$$

in which $p_{noise}$ is the total error probability, $\rho_{ideal}$ is the ideal density matrix, 1 is the identity matrix and $n$ is the number of qubits. We decompose $p_{noise}$ into the different error contributions and write

$$p_{noise} = 1 - (1 - p_P)^{N_P}(1 - p_R)^{N_R}(1 - p_F)^{N_F}. \qquad (28)$$

Here $p_P$ denotes the probability of spontaneous scattering during photon emission, $p_R$ the error probability during a Raman rotation, $p_F$ the error probability for the fusion process and $N_P$, $N_R$ and $N_F$ are the respective number of operations in the protocol. Note that we do not include mechanisms such as decoherence or qubit leakage in the above formula, as we are unable to assign a value to a specific step of the protocol.

In Extended Data Table 2, we compare the fidelity model to the measured lower and upper bounds as defined by equation (20) and equation (26), respectively. For the tree and box graph states, the predicted fidelities $\mathcal{F}_{model}$ are found to fall between the measured bounds, as expected. For the hexagon graph, $\mathcal{F}_{model}$ falls slightly above the upper bound but is still consistent with it when taking into account the statistical uncertainty (less than one standard deviation). As mentioned earlier, the model does not include the effect of qubit leakage, decoherence and drifts of, for instance, the magnetic field or optical fibres. Hence, it is likely that the predicted fidelities are slightly overestimated.

## Data availability

The datasets generated and/or analysed during this study are available at https://doi.org/10.5281/zenodo.10717770 (ref. 46). Source data are provided with this paper.

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

**Acknowledgements** P.T. thanks G. Styliaris for valuable discussions. This work was supported by the Bundesministerium für Bildung und Forschung through the Verbund QR.X (16KISQ019), by the Deutsche Forschungsgemeinschaft under Germany's Excellence Strategy - EXC-2111 - 390814868 and by the European Union's Horizon Europe research and innovation programme through the project QIA-Phase 1 under grant agreement no. 101102140. The opinions expressed in this document reflect only the author's view and reflects in no way the European Commission's opinions. The European Commission is not responsible for any use that may be made of the information it contains.

**Author contributions** P.T. and L.R. performed the experiments. P.T. and O.M. analysed the data. G.R. and O.M. supervised the project. All authors were involved in the discussion of the results and writing of the manuscript.

**Funding** Open access funding provided by Max Planck Society.

**Competing interests** The authors declare no competing interests.

**Additional information**
**Correspondence and requests for materials** should be addressed to Olivier Morin.

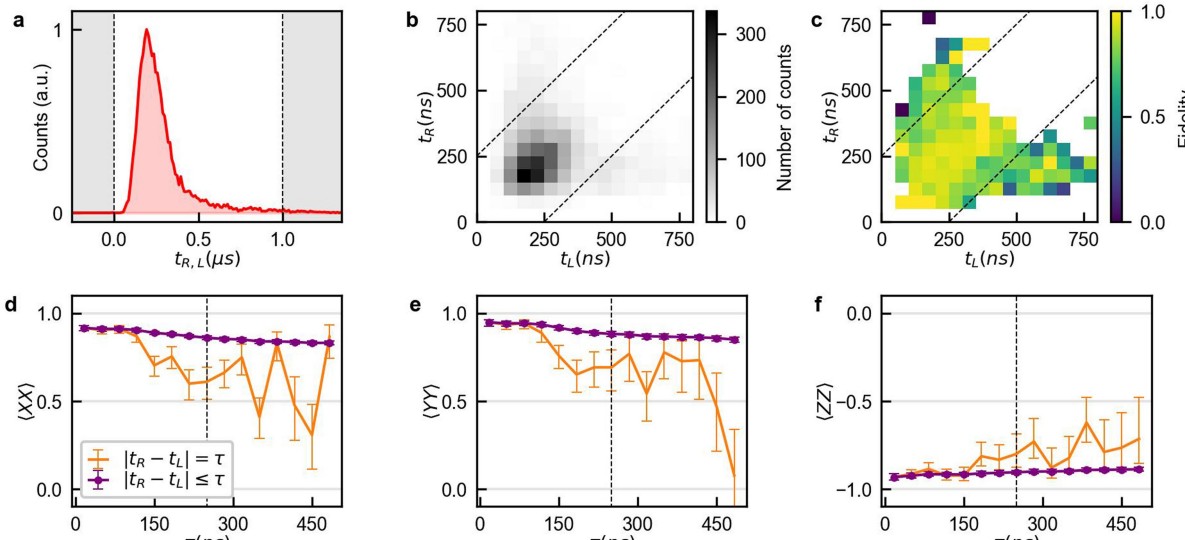

**Extended Data Fig. 1 | Atom–atom entanglement by means of the cavity-assisted fusion gate. a**, Histogram showing the total photon count rate as a function of $t_R$ and $t_L$, in which $t_{R/L}$ is the arrival time of the right-hand-polarized/left-hand-polarized photon generated in the fusion process. Only events in which both photons were detected are shown. Dashed lines mark the acceptance window for post-selection. **b**, Density plot of the number of counts as a function of $t_R$ and $t_L$. **c**, Density plot of the fidelity as a function of $t_R$ and $t_L$. **d–f**, Expectation values of the correlators $XX$, $YY$ and $ZZ$ as a function of photon arrival time difference. The orange line shows the correlator for time difference $|t_R - t_L| = \tau$, whereas the purple line is the cumulative correlator, meaning for events in which $|t_R - t_L| \leq \tau$. The dashed lines mark the maximum $\tau$ we choose for Fig. 1c. Error bars represent the $1\sigma$ standard error.

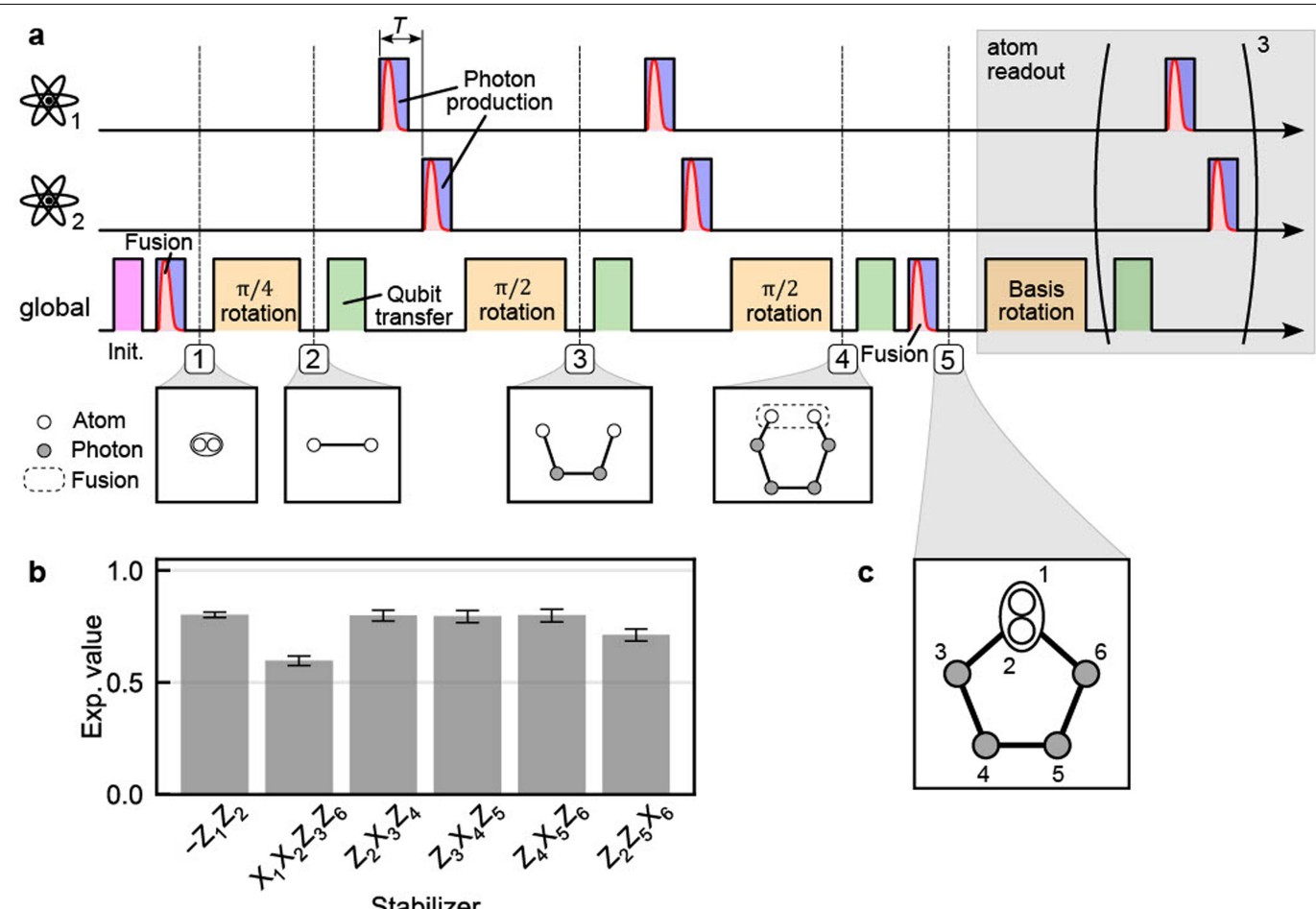

**Extended Data Fig. 2 | Protocol for the generation of the pentagon graph.** A two-atom graph state is obtained from the cavity-assisted fusion gate, followed by a −π/4 pulse. A chain is grown along one dimension using photon emissions and π/2 rotations on the atomic qubits. Both ends of the chain are merged to form a ring. Error bars represent the 1σ standard error.

**Extended Data Table 1 | Coincidence rate**

|  | Box | Pentagon | Hexagon | Tree |
|---|---|---|---|---|
| Number of events | 628 | 392 | 219 | 227 |
| Measurement time (h) | 4.5 | 3.3 | 8.7 | 4.8 |
| Rate (events/min) | 2.3 | 2.0 | 0.4 | 0.8 |

Coincidence-rate statistics for the generated box, pentagon, hexagon and tree graph states.

**Extended Data Table 2 | Error estimation**

|  | Box | Hexagon | Tree |
|---|---|---|---|
| $N_P$ | 6 | 8 | 8 |
| $N_R$ | 6 | 8 | 4 |
| $N_F$ | 2 | 2 | 1 |
| $\mathcal{F}_+$ | $0.80 \pm 0.02$ | $0.67 \pm 0.03$ | $0.85^{+0.02}_{-0.03}$ |
| $\mathcal{F}_{\mathrm{model}}$ | 0.74 | 0.69 | 0.77 |
| $\mathcal{F}_-$ | $0.59 \pm 0.03$ | $0.34^{+0.06}_{-0.07}$ | $0.69^{+0.04}_{-0.05}$ |

We compare the predicted fidelities with the measured upper and lower bounds of the fidelity. For our model, we use the following error probabilities for the different steps: $p_P$=0.02, $p_R$=0.01 and $p_F$=0.06.