## [Peer Review File · Nature]

Manuscript Title: Fusion of deterministically generated photonic graph states

Editorial Notes:

Reviewer Comments & Author Rebuttals

Reviewer Reports on the Initial Version:

Referees' comments:

Referee #1 (Remarks to the Author):

In this manuscript, Thomas et al demonstrate the fusion of graph states generated deterministically by two atomic quantum emitters in the same cavity. They use a fusion-like operation to redundantly encode the emitters (in a dual-rail type state) and then they pump them and collect the emitted photons. The types of states they create are ring graphs and tree graphs, which are states that cannot be generated by a single emitter.

I find this paper a milestone demonstration in the field of photonic quantum information processing. The authors build on their previous work, which recently appeared in Nature, where they showed record size graph states generated by pumping a single quantum emitter. Their present work takes the natural next step and generalizes this scheme to two emitters. While the fidelities are still not where we need them to be for practical applications, the experiment is an important step, and I believe the work should be suitable for publication in Nature once the authors address some questions/issues listed below.

1. I believe the title is a bit confusing, as “deterministic quantum emitters” is not quite what is meant here. I would suggest the alternative title “Fusion of photonic graph states generated deterministically from quantum emitters”, which clearly conveys that the deterministic element is the generation and not the emitters themselves.

2. The authors initialize the emitters in the “triplet zero” (or Ψ^+ Bell) state through the fusion process. How come this is the only state that is generated (or post selected?) and there is no singlet (or Ψ^- Bell) state also? Wouldn't the probabilistic nature of the photon measurement give either of the two?

3. The strategy used here based on the fusion can only generate certain types of states. If the two emitters were controlled through entangling unitary two-qubit gates, more general states could be generated. I believe this distinction merits a discussion.

4. The emitters are detuned from the cavity to suppress their interaction, but there must be some residual interaction since they are identical. What is the timescale of this interaction?

5. To give some intuition to the readers about the additional error magnitudes here compared to

single emitter protocols, if a linear cluster state was generated using two emitters (vs with a single emitter as in prior work) what would be the relative fidelity for a fixed size graph?

6. My understanding is that the main error comes from the imperfect interference of the photons, which in turn originates from the variance in arrival time. Is there a way to engineer the device to make this tighter and reduce the error?

7. What are the prospects of using the cavity to implement a two-qubit gate and avoid fusion?

8. The projector operator right above Eq. 2 does not seem to be consistent with the preparation of the triplet zero state (see question 2 above).

9. The procedure of obtaining tree graphs based on fusion limits the trees to specific depth/branching ratios. This point should be discussed.

10. In the Methods section titled "Post selection criteria" it is mentioned that "most photons are generated from $|F=2, m_F = \pm 2\rangle$ ". This is somewhat confusing because, if I understood correctly, all photons that are part of the final graph state are generated through spontaneous emission from this state. It may help to clarify this point.

11. Equation 7 seems to have issues. For one, there is only one "S" subscript, while there are two emitters. The authors should fix this and check all the equations in the Methods section.

Referee #3 (Remarks to the Author):

The paper reports on linear-optics fusion of multi-photon entangled states generated by two single atoms trapped in the same cavity. Up to 8 qubits are demonstrated in an impressive proof-of-concept demonstration of the capabilities of the platform. Various entangled states are synthesized and the state quality is quantified by measuring the stabilizer generators in order to set lower bounds on the state fidelity. For some of the states, a fidelity violating the classical bound is reported.

This is a truly impressive experimental work. However, the work may to some extent be considered a natural extension of the equally impressive work from the same group of 1D multi-photon entanglement generation published in Nature 2022 (ref. 16). Yet the present work contains clear novelties by operating two atoms and implementing fusion on the indistinguishable photons emitted from either atom. The natural question is, whether the present work constitutes such an important step forward compared to the 2022 paper that it warrants publication in Nature or rather would be better suited for a more specialized journal. I tend to lean towards the latter option and would consider the work more appropriate for a more specialized journal like, e.g., Nature Photonics. Indeed the fusion operations (referred to as cavity assisted fusion gates) are the same linear photonic operations reported by other groups working, e.g., with spontaneous parametric down-conversion sources or quantum dot sources. Also two-atom cavity QED systems have been reported before. Combining these functionalities is certainly new, but probably not the conceptual leap

warranting publication in Nature given the previous works.

A couple of more specific comments that I would encourage the authors to consider in a revision:

- 1) Scalability of the approach: the authors mention in the abstract that the approach is "in principle scalable" but give no further justification or quantification of that statement. I think it would be highly interesting to elaborate this point in some detail including the role of infidelities of the various operations and also account for the operation speed of the source.
- 2) The values of the fidelity are extracted from the measurements but essentially no insight is given as to what limits these values. A proper theoretical modeling of the error contributions and comparison to experimental values would give the reader a much better opportunity to judge the viability of the chosen approach.
- 3) the generation of the Bell pair is described in good detail, but the subsequent more elaborate fusion protocols are described in a rather rudimentary fashion. One option for improving the readability of the manuscript would be to elaborate further on the implementation of the protocol illustrated in Fig. 2a

Referee #4 (Remarks to the Author):

The manuscript describes the experimental generation of tree- and ring-type graphs of photonic and atomic qubits using two atoms in a single high-q cavity.

This is an expansion of a previous work by the same group (Nature 608, 677 (2022), here ref 16), in which a single trapped atom in a single-sided Fabri-Perot resonator was used to generate linear- and star- (GHZ) shaped photonic graphs.

Here the authors use two individually-addressed atoms, trapped in two different anti-nodes of the same resonator, to generate photonic graphs by the same protocols demonstrated in ref 16.

The key tool in this work is a photonic measurement that can probabilistically (50%) herald the preparation of the spin states of the two atoms into a Bell state (PRL 111, 100505 (2013), PRL 126, 230506 (2021), here refs 22,23). The two atoms are prepared in a product state, and measuring two photons at opposite polarizations projects them into the $01, 10$ subspace, in which they are at the $01+10$ Bell state.

Performing this procedure at different steps enables "fusing" the graphs generated from the two atoms into geometries like tree or ring, with the connecting parts being the "fused" two-atom state (in the ring case we have both atomic and photonic fused states) that can be viewed as a redundantly encoded single qubit - which justifies the term "fusion of graphs".

This is an impressive cavity-QED demonstration with high potential impact on an important field (photonic graph states are crucial both for photonic quantum computation, as well as for repeaters), and the results are of high quality.

Accordingly, in my view it is certainly a manuscript suitable for publication in Nature.

Having said that, there are a number of points that in my view need to be clarified, and comments that I hope the authors would consider.

1) The first point that needs further clarification is the motivation: the entanglement of the two atoms is in fact performed in a way that in some sense is equivalent to linear-optics fusion: two photons are emitted from the two atoms, and at 50% success rate the atoms are projected to a "fused" state.

Therefore, one can ask what is the advantage of this method compared to just fusing the two photonic graphs with linear optics (at 50% success rate as well)?

This could be done by having each atom (not two in the same cavity, but each in its own cavity like in Ref 16) just generate one extra photon, these two photons can be sent to a beam splitter for linear-optics fusion, and then the atoms are measured on some basis to disentangle them.

Even more efficiently, one could use the photon-atom SWAP gate to map the atomic states to photons that will be sent directly to linear-optics fusion, thereby performing the photon generation and the atom disentangling in a single step.

If the authors could discuss the advantages of their scheme compared to this alternative I think the motivation will be clearer - why is it better to rely on an atomic fused state?

One possible advantage I can think of is that this scheme is resilient to photon loss in the fusion stage - the probability of success is still 50%, but the atoms can be read multiple times.

Another possible advantage is that it results not in a solely photonic graph, but rather in a photonic-atomic graph state - which I assume is mostly relevant for repeater purposes?

In any case, a short discussion on this subject seems in place.

2) If the entangling of the atoms is the first stage it can be done in a repeat-until-success manner, which is good. However, when the entangling is performed in the middle or end of the graph generation process (as in the cases of both the tree and the ring), I believe a failure (in 50%) would mean the entire graph state would have to be discarded (like with regular fusion) - if this is correct, it needs to be mentioned.

3) This is a (very challenging) cavity-QED work - so on top of perhaps relating to this field in the introduction, some relevant physical (and technical) details should be mentioned (or mentioned more clearly) in the text:

* Only the cooperativity C is mentioned - but knowing the other parameters is important as well - what is the photon-atom coupling rate (for the cycling transition or for the specific transitions used) in MHz? What is the linewidth of the cavity, or more specifically K_i , K_{ex} (in MHz, not ppm of the

mirrors)?

* What is the overall transmission per photon? Namely, what is the probability to get a "click" at the detector after each vSTIRAP?

* What is the entire protocol time (for example for the 6,8 ring graphs)? I noticed 20 microseconds between vSTIRAPs but did not find the rest of the durations.

4) The errors in this manuscript are in the 10s of percentages but there's only one sentence (in the Methods) that refers to the imperfections that lead to the infidelity in the atom-atom fusion. I am missing a discussion on the sources of error and how they are expected to be reduced (higher C? Better Raman rotations?) such that these schemes are suitable for the applications mentioned.

5) On all the figures that present the expectation value for the stabilizers, there's no point in having y-axis ticks [0,0.5,1]. The reader can get more information from a smaller range as in ref 16.

6) Why is there such a high variance in the expectation values of the stabilizers? (maybe the encoded vertices of the graph?). In ref 16 the 15-qubit linear cluster state had stabilizer expectation values from 0.9 to ~ 0.96 , while here they vary from ~ 0.6 to ~ 0.9 .

7) The explanation of the phase in line 153 is a little unclear. The free evolution under the external magnetic field gives rise to a time-dependent phase $\exp(i*2*\pi*(2*100\text{kHz})*t_0)$ between the states of the atomic qubit, $1,-1$ and $1,+1$. So do I understand correctly that in order to sync the phases of the two branches of the tree before the atomic fusion, the authors adjust the wait time T between photon generations from the two atoms?

8) Denote free-evolution time (t_0) in the appropriate place in Fig3a.

Sincerely,

Prof. Barak Dayan

Weizmann Institute of Science

We thank all three Referees for their careful reading of the manuscript and their thorough reports. Below we respond to each of their remarks and suggestions, which we also address in the revised version of our manuscript. Changes in the text are highlighted in colour (red for removed parts, blue for added parts). We hope that the Referees find our manuscript to be ready for publication.

RESPONSE TO REFEREE #1

We thank the Referee for his/her in-depth report and valuable remarks. We appreciate that our work is acknowledged as a “milestone demonstration in the field of photonic quantum information processing”. We are glad that the Referee supports publication in Nature.

Here is our point-to-point reply:

1. I believe the title is a bit confusing, as “deterministic quantum emitters” is not quite what is meant here. I would suggest the alternative title “Fusion of photonic graph states generated deterministically from quantum emitters”, which clearly conveys that the deterministic element is the generation and not the emitters themselves.

We agree that the word “deterministic” should refer to the generation process, not the emitters. Our phrase “deterministic emitters” is community slang to distinguish our emitters from probabilistic sources such as those based on parametric down conversion. In order to avoid confusion and better highlight our achievement, we have changed the title to “Fusion of deterministically generated photonic graph states”.

We hope the referee likes this change.

2. The authors initialise the emitters in the “triplet zero” (or Ψ^+ Bell) state through the fusion process. How come this is the only state that is generated (or post selected?) and there is no singlet (or Ψ^- Bell) state also? Wouldn't the probabilistic nature of the photon measurement give either of the two?

The answer to this question is rooted in the fact that the cavity has only one output mode. In a standard linear optical Bell measurement setting a beam splitter with two output modes is used. Therefore, the two photons can either leave through the same or different output modes of the beam splitter. This determines whether the spatial component of the projected wavefunction is symmetric or antisymmetric. Due to bosonic symmetry properties this immediately determines the symmetry of the spin wavefunction as well. In our cavity, where only one output mode is available, the two photons can only occupy a symmetric spatial wavefunction. Consequently, the projection results in the symmetric “triplet zero” $|\Psi^+\rangle$ state. We have added a short mathematical derivation of this process in the Methods. This result is also in agreement with the findings of Casabone *et al.*, PRL **111**, 100505 (2013) (Ref. 22) and Langenfeld *et al.*, PRL **126**, 230506 (2021) (Ref. 23).

3. The strategy used here based on the fusion can only generate certain types of states. If the two emitters were controlled through entangling unitary two-qubit gates, more general states could be generated. I believe this distinction merits a discussion.

This point seems closely related to question 9. We therefore discuss both points below.

4. The emitters are detuned from the cavity to suppress their interaction, but there must be some residual interaction since they are identical. What is the timescale of this interaction?

The interaction of the two atoms via the cavity field is indeed not zero, but scales as $1/\Delta^2$, with Δ being the single photon detuning (neglecting other interactions such as the van der Waals interaction which does not play a role for our atomic distances). We have characterised this in a previous work (Ref. 36) and find the effect to be on the order of 10^{-3} per photon, much smaller than other error sources.

5. To give some intuition to the readers about the additional error magnitudes here compared to single emitter protocols, if a linear cluster state was generated using two emitters (vs with a single emitter as in prior work) what would be the relative fidelity for a fixed size graph?

In the Methods, we have added a section on the different sources of error in our system. The simple model we introduced now gives an estimate of the fidelity of a given graph taking into account the contributions of different steps in the protocol. In the case of a linear cluster state, the fidelity is expected to be lower in the two-emitter case by a factor corresponding to the fidelity of the fusion gate.

6. My understanding is that the main error comes from the imperfect interference of the photons, which in turn originates from the variance in arrival time. Is there a way to engineer the device to make this tighter and reduce the error?

The imperfect interference does not stem from the non-zero temporal width of the wavefunction (i.e. variance in arrival time), but from photon distinguishability. If the photons were perfectly indistinguishable in the temporal degree of freedom, we would expect unity fidelity for the fusion gate, even if both photons are detected with a relative time delay given by the duration of the single-photon wave packet.

The main sources of photon distinguishability in this process are spontaneous scattering from the excited state, stemming from a finite cooperativity C , and a shot-to-shot variation in atom positions. The latter leads to slightly varying coupling rates to the cavity field and the control laser. A way to fix this would be the implementation of optical tweezers for more reliable and precise positioning. In both cases though, the effect scales with the pulse area of the photon, not its length.

7. What are the prospects of using the cavity to implement a two-qubit gate and avoid fusion?

This is a very relevant remark as such a gate mechanism has been demonstrated (Welte *et al.*, PRX **8**, 011018 (2018)), involving an ancillary photon that reflects off the cavity. The biggest hurdle on this route is the need for an on-demand single-photon source with a high enough efficiency. In past implementations attenuated laser pulses were used in place of a single-photon source. However, this always results in a compromise between efficiency and fidelity, mainly due to the two-photon contribution in the coherent state input. Given the current capabilities of our system, the fusion is still the overall better compromise. On the long run, if the gate mechanism can be improved (proper single photon source, better

cooperativity, better escape efficiency), the two-qubit gate is certainly the preferred choice.

8. The projector operator right above Eq. 2 does not seem to be consistent with the preparation of the triplet zero state (see question 2 above).

This is correct. For the preparation of the triplet zero state, the atoms are initialised in $|F = 2, m_F = 0\rangle$ and therefore do not carry qubits. Hence, the projector above Eq. 2 does not apply. We hope to have clarified this point in our answer to question 2.

9. The procedure of obtaining tree graphs based on fusion limits the trees to specific depth/branching ratios. This point should be discussed.

We don't fully understand the remark. The depth and branching ratio of the tree graph is not limited by the fusion itself, but by the number of emitters (or memories), as is the case for unitary two-qubit gates. It may be the case that for a certain depth/branching ratio fewer emitters and/or entangling operations are needed when using a two-qubit unitary. However, it is typically not a trivial problem to find the minimum resources required for a certain graph. Indeed, as shown by S. Economou and coworkers (Ref. 18, 19, 20) the optimal algorithm is usually not straightforward.

We already mentioned in the conclusion/outlook of the manuscript that to generate a tree of larger depth, more emitters are necessary. We believe that a more detailed discussion is beyond the scope of this manuscript.

This also relates to question 3. that was brought forward by the Referee. We would argue that our protocol enables the generation of arbitrary graphs, given a sufficient number of emitters is available (see for instance Ref. 19). The way in which we carry out the fusion may differ slightly from a unitary CPHASE gate in the way vertices connect in the process. However, this can be compensated by additional emitters and redundant encoding.

10. In the Methods section titled "Post selection criteria" it is mentioned that "most photons are generated from $|F = 2, m_F = \pm 2\rangle$ ". This is somewhat confusing because, if I understood correctly, all photons that are part of the final graph state are generated through spontaneous emission from this state. It may help to clarify this point.

The photons are not generated by spontaneous emission but by vacuum STIRAP. This is why the starting state is a ground state and not an excited state. The statement that all photons that are part of the final graph state are generated from $|F = 2, m_F = \pm 2\rangle$ is correct only for the ring state protocol, where in the first step the photons are projected in order to entangle the emitters. For the tree states, however, the first two photons are generated from $|F = 2, m_F = 0\rangle$, while also being part of the final graph state. So indeed, the statement only applies to "most photons". In order to avoid confusion we have slightly modified the mentioned paragraph in the Methods section.

11. Equation 7 seems to have issues. For one, there is only one "S" subscript, while there are two emitters. The authors should fix this and check all the equations in the Methods section.

This is indeed correct. We revised the manuscript accordingly.

RESPONSE TO REFEREE #3

We thank the Referee for his/her report. We appreciate that the Referee considers our work as “truly impressive”. While the Referee acknowledges clear novelties compared to our previous experimental work, they raise the question whether the present submission contains enough novelties for a journal like Nature. We acknowledge that this is a legitimate question to ask and answer as follows:

We agree with the Referee that fusion of photonic graph states has been demonstrated in a number of experiments, but to the best of our knowledge only using the probabilistic and rather inefficient spontaneous parametric down-conversion sources. To overcome the limitations of these sources, a growing number of theoretical as well as experimental studies investigate ways to generate photonic graph states deterministically from individual photon emitters. Since a single emitter is not enough to generate arbitrary graphs, the entire approach depends on the ability to merge graphs from multiple emitters. On a fundamental level, this is the last crucial step towards tailoring the entanglement topology at will.

We now report on this decisive step in our manuscript and therefore believe that our work has groundbreaking implications for the field of photonic quantum computing, as also pointed out by Referees #1 and #4. We further believe that the fusion mechanism enabled by the cavity extends the general concept of fusion-based quantum computing. Moreover, our work is in principle transferable to various other platforms based on cavity QED, e.g. ions, colour centres, etc.

Here is our point-to-point reply:

1. Scalability of the approach: the authors mention in the abstract that the approach is “in principle scalable” but give no further justification or quantification of that statement. I think it would be highly interesting to elaborate this point in some detail including the role of infidelities of the various operations and also account for the operation speed of the source.

Apart from the abstract, we do briefly discuss this point in the conclusion of the main text. However, the Referee is correct in that the level of detail is kept rather low. In our view, the quantitative projection of the system performance with many emitters is a work on its own and depends on many different variables. Such a discussion would soon become highly technical and would quickly exceed the scope of this manuscript. The main aspect we wanted to highlight with “in principle scalable” is that the size and complexity of the generated graph state can be increased by adding more emitters. For this, the crucial ingredient is the ability to entangle the emitters via the cavity-assisted fusion gate, which we discuss in good detail. To further stress this point, we have now added a discussion concerning the possibilities of scaling up the number of emitter qubits in the conclusion.

2. The values of the fidelity are extracted from the measurements but essentially no insight is given as to what limits these values. A proper theoretical modelling of the error contributions and comparison to experimental values would give the reader a much better opportunity to

judge the viability of the chosen approach.

Firstly, we would like to stress that we did not measure the fidelity, but the expectation value of an operator that puts a lower bound on the fidelity. The intrinsic uncertainty of the obtained quantity alone makes a comparison between theory and experiment difficult. This being said, understanding how different error sources affect the fidelity is indeed crucial in order to improve the system for future experiments, but it becomes a difficult task as systems are getting more complex and the sources of errors smaller. This is particularly the case in the context of multi-qubit graph states, since the effect of errors depends on the entanglement topology in a non-trivial way (see Ref. 43, Methods).

We have to some extent quantified the errors in our previous work (Ref. 16). The fusion process presents itself as the main additional error source in the present work. While presumably its fidelity is predominantly limited by photon distinguishability, the quantitative modelling of this process alone is a challenging problem due to the large Hilbert space of temporal modes. We are not aware of any theoretical framework that accurately describes two emitters simultaneously emitting into the same cavity mode, while taking into account the temporal degree of freedom of the photons.

Furthermore, we are currently in the process of investigating an error mechanism caused by a qubit leakage into states that would normally not be occupied at any point in the protocol (those are $|F = 1, m_F = 0\rangle$ and $|F = 2, m_F = \pm 1\rangle$). These are detrimental to the fidelity as they can take part in the Raman transfers and photon emissions. Due to the complex level structure of the atom this mechanism is again extremely difficult to investigate quantitatively with a satisfying accuracy, mainly due to the lack of an m_F -selective state readout.

Since all Referees have raised this point in one way or another, we realise that this topic deserves closer attention. We have therefore added a discussion in the main text and a more detailed section in the Methods. The latter includes a simple model that takes into account the most important error mechanisms. Moreover, we have derived an upper bound for the fidelity, in addition to the lower bound. The two bounds define an interval that puts a tighter constraint on the possible values for the fidelity. A comparison between this interval and the value predicted by our theoretical model is made in the Methods.

3. the generation of the Bell pair is described in good detail, but the subsequent more elaborate fusion protocols are described in a rather rudimentary fashion. One option for improving the readability of the manuscript would be to elaborate further on the implementation of the protocol illustrated in Fig. 2a

We agree with the Referee that the description of the protocol is kept rather short, which is partly owed to the fact that some of the steps are the same as in Ref. 16. For self-consistency and to improve readability, we have followed the Referee's advice and added more experimental detail to the protocol description corresponding to Fig. 2a.

RESPONSE TO REFEREE #4

We thank the Referee for his/her positive evaluation of our manuscript and many useful comments. We are delighted that the Referee considers our work to be “an impressive cavity-QED demonstration with high potential impact”.

Here is our point-to-point reply:

1. The first point that needs further clarification is the motivation: the entanglement of the two atoms is in fact performed in a way that in some sense is equivalent to linear-optics fusion: two photons are emitted from the two atoms, and at 50% success rate the atoms are projected to a “fused” state.

Therefore, one can ask what is the advantage of this method compared to just fusing the two photonic graphs with linear optics (at 50% success rate as well)?

This could be done by having each atom (not two in the same cavity, but each in its own cavity like in Ref 16) just generate one extra photon, these two photons can be sent to a beam splitter for linear-optics fusion, and then the atoms are measured on some basis to disentangle them.

All of the above is correct. Both fusion mechanisms are conceptually very similar, but have their strengths and weaknesses regarding their practical implementation:

The advantage of using one cavity is the perspective of scaling up to more atoms with little additional hardware overhead. Moreover, the photon indistinguishability is almost intrinsically guaranteed, as both atoms couple to the same cavity and control laser.

Another advantage is the programmability of the system. Indeed, with more atoms in the same cavity, one can apply the fusion to an arbitrary pair of emitters at any step of a protocol. An architecture of multiple atom-cavity systems requires efficient and high-fidelity photon routing, which for polarisation qubits is challenging, both in free-space and with integrated photonics.

Even more efficiently, one could use the photon-atom SWAP gate to map the atomic states to photons that will be sent directly to linear-optics fusion, thereby performing the photon generation and the atom disentangling in a single step.

If the authors could discuss the advantages of their scheme compared to this alternative I think the motivation will be clearer - why is it better to rely on an atomic fused state?

One possible advantage I can think of is that this scheme is resilient to photon loss in the fusion stage - the probability of success is still 50%, but the atoms can be read multiple times.

Indeed, as mentioned in the Methods section, the readout of the atoms can be repeated several times (in fact, independent of the fusion). In the presence of photon loss, this is more efficient than a photon measurement.

Another possible advantage is that it results not in a solely photonic graph, but rather in a photonic-atomic graph state - which I assume is mostly relevant for repeater purposes?

This is correct. In some cases, it is required that the emitter is part of the graph. For example, in the one-way repeater protocol proposed by Borregaard *et al.*, PRX **10**, 021071 (2020), the emitter qubit forms the root qubit of the tree graph state.

In any case, a short discussion on this subject seems in place.

We added these explanations in the main text.

2. If the entangling of the atoms is the first stage it can be done in a repeat-until-success manner, which is good. However, when the entangling is performed in the middle or end of the graph generation process (as in the cases of both the tree and the ring), I believe a failure (in 50%) would mean the entire graph state would have to be discarded (like with regular fusion) - if this is correct, it needs to be mentioned.

It is indeed correct that a failure of the fusion implies a failure of the ring/tree generation protocol. However, as long as both photons are detected (i.e. RR or LL), the portion of the graph generated before the fusion can still be recovered. The entire graph state has to be discarded only if one or both photons are lost in the fusion process. We have added an explanation to the main text and thank the referee for this valuable comment.

3. This is a (very challenging) cavity-QED work - so on top of perhaps relating to this field in the introduction, some relevant physical (and technical) details should be mentioned (or mentioned more clearly) in the text:

- Only the cooperativity C is mentioned - but knowing the other parameters is important as well - what is the photon-atom coupling rate (for the cycling transition or for the specific transitions used) in MHz? What is the linewidth of the cavity, or more specifically K_i , K_{ex} (in MHz, not ppm of the mirrors)?

We have added the relevant cavity QED parameters to the main text. Note that the value of C slightly changed. This is because, for the purpose of self-consistency, the coupling g now refers to the transition $|F = 1, m_F = \pm 1\rangle \leftrightarrow |F' = 2, m'_F = \pm 2\rangle$, which is the most relevant in the protocol.

- What is the overall transmission per photon? Namely, what is the probability to get a "click" at the detector after each vSTIRAP?

The probability that one of the detectors clicks after a given emission attempt is what we refer to in the main text as "overall source-to-detector efficiency". This number, which we recently improved to close to 50%, includes the source efficiency, transmission through the optical setup and detector efficiency.

- What is the entire protocol time (for example for the 6,8 ring graphs)? I noticed 20 microseconds between vSTIRAPs but did not find the rest of the durations.

We added some detail to the description of the ring state protocol as well as Fig. 2a, including the duration of certain pulses. These durations are the same for the tree state protocol. The full duration of the different protocols is now mentioned in the figure captions.

4. The errors in this manuscript are in the 10s of percentages but there's only one sentence (in the Methods) that refers to the imperfections that lead to the infidelity in the atom-atom fusion. I am missing a discussion on the sources of error and how they are expected to be reduced (higher C? Better Raman rotations?) such that these schemes are suitable for the applications mentioned.

This point was also raised by the other Referees. We have therefore devoted a section in the Methods to the discussion of the different sources of error and how the system can be improved.

5. On all the figures that present the expectation value for the stabilisers, there's no point in having y-axis ticks [0,0.5,1]. The reader can get more information from a smaller range as in ref 16.

We agree that a smaller range would be beneficial for some of the plots. However, as the displayed values vary much more than in Ref. 16, we decided to keep the scale uniform across all plots for the purpose of clarity.

6. Why is there such a high variance in the expectation values of the stabilisers? (maybe the encoded vertices of the graph?). In ref 16 the 15-qubit linear cluster state had stabiliser expectation values from 0.9 to ~ 0.96 , while here they vary from ~ 0.6 to ~ 0.9 .

In Ref. 16 all qubits are equivalent in the sense that the protocol is identical for all qubits with index $k > 1$. For the two-emitter protocols presented in this work this is not the case and we have an additional error mechanism given by the fusion. Stabilisers corresponding to qubits located at or adjacent to a fused vertex, thereby experience a larger error, as correctly suggested by the referee. This is the case for most stabilisers, with the exception of the qubits on the lowest layer of the tree graph.

7. The explanation of the phase in line 153 is a little unclear. The free evolution under the external magnetic field gives rise to a time-dependent phase $\exp(i*2*\pi*(2*100\text{kHz})*t_0)$ between the states of the atomic qubit, $1,-1$ and $1,+1$. So do I understand correctly that in order to sync the phases of the two branches of the tree before the atomic fusion, the authors adjust the wait time T between photon generations from the two atoms?

This is correct. Varying T is equivalent to shifting the $\pi/2$ pulse for the second atom, thereby adjusting the phase factor $\exp(i\phi(t_0))$ for the second branch. From the comment of the Referee we realised that Eq. 1 does not properly distinguish between the phases that correspond to the two atoms. We therefore added an expression for the phase $\phi_{1,2}$ distinguishing between the emitters, hoping to make the description of the mechanism more accessible.

8. Denote free-evolution time (t_0) in the appropriate place in Fig3a.

The parameter t_0 is now indicated in Fig. 3a.

We again thank all the Referees for their valuable remarks and hope that they consider our manuscript in its current form suitable for publication in Nature.

Reviewer Reports on the First Revision:

Referees' comments:

Referee #1 (Remarks to the Author):

The authors have addressed my comments and questions, as well as those of the other reviewers. I find this version of the manuscript much improved, and I believe that this work is a very good fit for Nature, considering it's the first demonstration of the fusion of deterministically generated graph states. I support acceptance in Nature without further changes.

Referee #4 (Remarks to the Author):

I find the manuscript significantly improved and recommend it for publication.

My only remaining reservation concerns the introduction, which currently lacks background coverage of the field of single-atom cavity-QED, other than the authors' own previous works.

Specifically, it would be appropriate to reference the seminal theoretical work by Duan and Kimble (PRL 92, 127902 (2004)), as well as to relate and compare to the approach of using gates to connect the already generated graphs, in contrast to fusion measurement.

In any case, I congratulate the authors on this very nice experimental demonstration.

Author Rebuttals to First Revision:**RESPONSE TO REFEREE #1**

We are happy to learn that we were able to resolve all comments brought forward by the referee, and thank him/her for recommending publication in *Nature* in its present form.

RESPONSE TO REFEREE #4

We thank the referee for his/her kind words of appreciation (*"In any case, I congratulate the authors on this very nice experimental demonstration"*) and the recommendation to publish our manuscript in *Nature* (*"I find the manuscript significantly improved and recommend it for publication"*).

The referee expressed a small reservation that we would like to address below:

Referee: "My only remaining reservation concerns the introduction, which currently lacks background coverage of the field of single-atom cavity-QED, other than the authors' own previous works.

Specifically, it would be appropriate to reference the seminal theoretical work by Duan and Kimble (PRL 92, 127902 (2004)), as well as to relate and compare to the approach of using gates to connect the already generated graphs, in contrast to fusion measurement."

We understand that the referee asks for a broader introduction into the field of single-atom cavity QED including the work of Duan and Kimble. We have therefore added the new reference 22 (Reiserer et al., Rev. Mod. Phys. 87, 1379 (2015)) which provides a broad introduction into the field and discusses both the proposal and the implementation of the Duan-Kimble scheme in great detail.

Connecting multiple sources with unitary gates as mentioned by the referee is indeed an interesting perspective. There are several ways to realise this in practice, with the Duan-Kimble scheme being only one of them. Another possibility could be to employ Rydberg-Rydberg interaction which has been implemented in numerous quantum-computing experiments of many groups around the world, but so far without a cavity. We feel that incorporating into our manuscript a discussion concerning the question which gate (or which fusion) is the best would require substantial changes to the main text, would significantly extend the length of the manuscript, would deviate from the straight story line of the manuscript, and would likely be incomplete or overwhelm the reader. In the interest of readability, we are therefore inclined to restrict ourselves to the implemented fusion gate and hope that the referee agrees.